# Clinical and molecular characteristics associated with high PD-L1 expression in EGFR-mutated lung adenocarcinoma

Jeremy Slomka[1], Hugo Berthou[2], Audrey Mansuet-Lupo[3,4,5], Hélène Blons[5,6,7], Elizabeth Fabre[2], Ivan Lerner[5,8], Bastien Rance[5,8], Marco Alifano[5,9], Jeanne Chapron[1], Gary Birsen[1], Laure Gibault[10], Jennifer Arrondeau[11], Karen Leroy[4,5,6], Marie Wislez[1,4,5] *

1 Thoracic Oncology Unit, Pneumology, Cochin Hospital AP-HP Paris, Paris, France, 2 Thoracic Oncology, Georges-Pompidou European Hospital, AP-HP Paris, Paris, France, 3 Pathology Department, Cochin Hospital, AP-HP Paris, Paris, France, 4 Team "Cancer, Immune Control and Escape" Inserm U1138, Cordeliers Research Centre, Paris, France, 5 Faculty of Medicine, Université Paris Cité, Paris, France, 6 Biochemistry Department, Molecular Oncology and Pharmacogenetics Unit, Georges-Pompidou European Hospital, AP-HP Paris, Paris, France, 7 Immunotherapy and Antiangiogenic Treatment in Cancerology, INSERM U970, Université Paris-Cité, Paris, France, 8 Informatics and Practice Evaluation, Georges-Pompidou European Hospital, AP-HP Paris, Paris, France, 9 Thoracic and Cardiovascular Surgery, Cochin Hospital, AP-HP Paris, Paris, France, 10 Pathology Department, Georges-Pompidou European Hospital, AP-HP Paris, Paris, France, 11 Medical Oncology, Cochin Hospital, AP-HP Paris, Paris, France

☯ These authors contributed equally to this work.
* marie.wislez@aphp.fr

**Data Availability Statement:** All relevant data are within the manuscript and its Supporting Information files.

## Abstract

### Objective

Recent evidence suggests that elevated levels of PD-L1 expression may be linked to early resistance to TKI and reduced survival in NSCLC with *EGFR* mutations. This study aimed to characterize the clinical and molecular features of *EGFR*-mutated lung adenocarcinomas and determine the prognostic significance associated with high PD-L1 expression.

### Materials and methods

We conducted a retrospective chart review of 103 consecutive patients with advanced *EGFR*-mutated NSCLC, who received treatment between 01/01/2016 and 30/12/2020, at our institution.

### Results

Among the tumors, 17% (n = 18) exhibited high PD-L1 expression (≥50% tumor proportion score), which was associated with a lower prevalence of common *EGFR* mutations (56% vs. 82%, p = 0.03) and a higher frequency of complex *EGFR* mutations (28% vs. 7%, p = 0.02). Univariate analysis did not reveal any significant differences in first-line response, progression-free survival, or overall survival between the PD-L1 ≥50% and <50% groups. However, multivariate analysis demonstrated that PD-L1 ≥50% was independently associated with shorter survival (HR = 2.57; 95%CI[1.20–5.55]; p = 0.02), along with male gender (HR

**Funding:** The author(s) received no specific funding for this work.

**Competing interests:** The authors have declared that no competing interests exist.

= 2.77; 95%CI[1.54–4.19]; p<0.005), presence of liver metastases (HR = 5.80; 95%CI [2.86–11.75]; p<0.005) or brain metastases (HR = 1.99; 95%CI[1.13–3.52]; p = 0.02), and poor general condition at diagnosis (ECOG 3 and 4) (HR = 10.69; 95% CI[4.42–25.85]; p<0.005). Additionally, a trend towards a higher frequency of *de novo* resistance was observed in the PD-L1 >50% group (7% vs. 17%, p = 0.19).

## Conclusion

High PD-L1 expression was more commonly found in lung adenocarcinomas with uncommon and complex *EGFR* mutations. Furthermore, high PD-L1 expression independently predicted poor survival. These findings warrant validation through prospective studies.

## Introduction

Lung cancer remains a significant health burden in France, accounting for a quarter of cancer-related deaths and ranking as the leading cause of cancer mortality [1]. The overall 5-year survival rate for lung cancer across all stages is merely 21%, with the majority of cases (57%) being diagnosed at an advanced stage [2]. In the pursuit of improved therapeutic outcomes, the concept of personalized medicine has gained prominence. Tailoring therapeutic strategies for advanced non-small-cell lung carcinoma (NSCLC) relies on identifying specific biomarkers, such as programmed death ligand 1 (PD-L1) expression in tumor cells and the presence of oncogenic addictions like epidermal growth factor receptor (*EGFR*) mutations.

*EGFR* mutations, a prevalent oncogenic driver in lung adenocarcinomas among the Caucasian population, occur in approximately 11–15% of cases and represent the most common targetable oncogenic addiction in the first-line setting [3]. These mutations can be classified into common mutations, rare mutations, and complex mutations [4]. Rare mutations encompass a heterogeneous group of uncommon point mutations (L861Q, G719X, etc.), in-phase insertion mutations in exon 20, and exceedingly rare mutations. Complex mutations involve the combination of multiple *EGFR* mutations, typically affecting different exons, including or excluding the exon 19 deletion or the L858R missense mutation of exon 21, and account for less than 3% of cases. Each mutation variant exhibits varying sensitivity to different EGFR-tyrosine kinase inhibitors (TKIs) [5]. Osimertinib, a third-generation TKI, is the standard first-line treatment for sensitive *EGFR* mutations [6]. In cases of TKI progression without identifiable targetable resistance mechanisms, platinum-based chemotherapy remains the reference treatment, often combined with a specific VEGF inhibitor [7].

Although immune checkpoint inhibitors alone have not shown improved overall survival compared to chemotherapy in patients with *EGFR*-mutated advanced NSCLC [8], the significance of high PD-L1 expression levels has been increasingly recognized. High PD-L1 expression, defined as staining in 50% or more tumor cells on immunohistochemistry, is significantly less prevalent in *EGFR*-mutated tumors (15.5%) compared to *EGFR* wild-type tumors (31%) [9]. Recent evidence has suggested that high PD-L1 expression may serve as a prognostic marker during TKI treatment, with worse outcomes observed in patients with elevated PD-L1 expression [10–15]. Therefore, the objective of this study was to analyze the clinical and molecular characteristics of a retrospective cohort of patients with *EGFR*-mutated NSCLC, who received treatment for metastatic disease between January 1, 2016, and December 30, 2022, at our institution. The aim was to characterize *EGFR*-mutated NSCLC with PD-L1 $\geq$ 50% in order to assess whether this subgroup has a distinct prognosis.

## Materials and methods

### Study design and study population

We conducted a multicenter retrospective observational cohort study encompassing patients diagnosed and followed up for *EGFR*-mutated advanced non-small cell lung carcinoma (NSCLC) at Cochin or European Georges Pompidou Hospitals in Paris, France. The inclusion criteria consisted of patients aged over 18 years, with *EGFR*-mutated advanced NSCLC, with histological confirmation, availability of PD-L1 expression analysis by immunohistochemistry, molecular analyses by next-generation sequencing performed as part of routine care between 01/01/2016 and 30/12/2020. Living patients who met the inclusion criteria were provided with an information letter and a clinical research non-objection form (see appendix). The study was ethically approved by the CERAPHP. Centre ethics committee on June 3, 2021 (reference 2021_06_10).

### Data flow

Data extraction from laboratory files covering the period between January 1, 2016, and December 30, 2020, allowed for the identification of patients with *EGFR*-mutated tumors. A manual review of patient records was then performed to identify individuals who met the inclusion criteria for the study. To ensure data confidentiality, the selected patients were anonymized before entering their information into the database. The data were accessed for research purposes from November 2021 to June 2022. Collected characteristics of the study population included demographic data, locations of secondary tumors, genetic alterations at diagnosis and progression, histological pathology characteristics, first-line metastatic treatment received, best response according to RECIST criteria 1.1, details of progression (date, site, and type), subsequent lines of treatment, and date of last news or death, with a last-point date of June 30, 2022 (basic demographic and clinical data are shown in S1 File). The median follow-up for living patients was 46.4 months.

### Evaluation criteria

The study had three primary objectives. The first objective was to characterize NSCLC with *EGFR* mutation and high expression of PD-L1. The second objective was to compare clinical responses based on PD-L1 status and assess its prognostic significance. The third objective was to evaluate changes in PD-L1 expression levels upon progression after the first-line treatment.

### Assessment of PD-L1 level and molecular characterization

PD-L1 Tumor Proportion Scores (TPS) were collected at the time of diagnosis and when available at relapse. The following antibodies were used at diagnosis: E1L3N (Cell Signaling Technology) for 38 tumors, 22C3 (Agilent) for 31 tumors, and QR1 (Diagomics) for 34 tumors.

 *EGFR* mutation and co-mutation status (specifically *TP53*, *CTNNB1*, *SMAD4*, or *PIK3CA*) were identified during routine care using next-generation sequencing. DNA was extracted from formalin-fixed paraffin-embedded tissues or liquid biopsies, and the analysis was performed using colonlung v1 or v2 Ampliseq panels developed by Thermofisher. The sequencing was conducted using IonTorrent sequencing chemistry on PGM, S5, or Proton IonSequencers from Thermofisher. These panels consisted of amplicons targeting hotspot mutations in *KRAS*, *EGFR*, *BRAF*, *PIK3CA*, *AKT1*, *ERBB2*, *PTEN*, *NRAS*, *STK11*, *MAP2K1*, *ALK*, *DDR2*, *CTNNB1*, *MET*, *TP53*, *SMAD4*, *FBX7*, *FGFR3*, *NOTCH1*, *ERBB4*, *FGFR1*, and *FGFR2* genes. *EGFR* mutation types were classified as common (exon 19 deletion and exon 21 L858R

**Table 1. EGFR complex mutations.**

| |
|---|
| G709A+ G719S |
| G719A + S768I (n = 2) |
| G719C + S768I |
| G719S + R776H |
| G719A + R776S |
| del19 + T790M |
| del19 + V769M |
| L858R + I744M + I715S |
| L858R + T790M |
| L861Q+ S768I |

List of complex EGFR mutations identified (n = 1 for each genotype except for G719A + S768I).

missense mutation), exon 20 insertion, complex (presence of at least two distinct *EGFR* mutations, Table 1), and rare (non-common, non-complex, non-insertion mutations in exon 20).

## Statistical analysis

We performed a comparative analysis between the populations of *EGFR*-mutated NSCLC with PD-L1 expression $\geq$ 50% and < 50% based on clinical, radiological, biological, and molecular criteria. Due to the limited number of patients in the highly expressed PD-L1 group (less than 20 patients), categorical variable analysis was conducted using a non-parametric Fisher exact test. Quantitative variables were analyzed using a non-parametric Mann-Whitney test. A p-value of $\leq$ 0.05 was considered statistically significant and were added to Table 2. We constructed Kaplan-Meier curves and employed a log-rank test to compare overall survival (OS) and progression-free survival (PFS). Univariate analysis for PFS and OS was performed using a Cox model. The variables included in the multivariate analysis, determined using a stepAIC method [16], were as follows: for PFS—presence of liver metastasis, brain metastasis, smoking, PD-L1 level, and presence of co-mutations; for OS—presence of brain and liver metastasis, general status, age, male sex, PD-L1 level, and presence of common mutations. Proportional hazards assumptions were verified. Since 49 out of the 103 patients in the study did not have LDH assessment at diagnosis, the univariate analysis of the study population could not include this variable. Nevertheless, we conducted a separate univariate analysis of the above-normal LDH level for the remaining 54 patients. An exploratory analysis of PFS and OS was conducted based on pack-years. Given missing data for 25 patients, this analysis, both univariate and multivariate, following the same modalities as previously described, was conducted separately on the population of 88 patients with available information. The results are shown in S1 Table.

## Results

### Baseline clinical and molecular characteristics

During the period from January 1, 2016, to June 30, 2020, a total of 394 patients with NSCLC were identified to have an *EGFR* mutation through NGS sequencing. Among them, 144 patients were excluded due to off-site follow-up, 108 due to localized disease, and 39 due to a lack of PD-L1 analysis. Ultimately, 103 patients were included in the study (Fig 1).

Among the 103 patients, 17% (n = 18) had a PD-L1 TPS $\geq$ 50% at diagnosis, while 83% (n = 85) had a PD-L1 TPS < 50%. There were no significant differences in mean age and gender distribution between the two groups. The proportion of smokers was higher in the

**Table 2. Baseline clinical characteristics of study participants according to PD-L1 expression.**

| Characteristic | PD-L1 < 50% | PD-L1 ≥ 50% | p-value |
|---|---|---|---|
| **All patients–number (%)** | 85 (82%) | 18 (17%) | |
| **Average age—year (standard deviation)** | 67.24 (13.63) | 67.50 (10.75) | 0.36 |
| **Sex** | | | |
| Female—number (%) | 57 (67%) | 11 (61%) | 0.78 |
| Male—number (%) | 28 (33%) | 7 (39%) | |
| **Smoking status** | | | |
| Never—number (%) | 49 (58%) | 7 (39%) | 0.19 |
| Current/Former—number (%) | 36 (42%) | 11 (61%) | |
| Number of packs years—average %* | 17.8 | 20.9 | 0.1 |
| Time between diagnosis and discontinuation—average in months | 25.9 | 15.9 | |
| **ECOG—number (%)** | | | |
| ≤2 | 78 (92%) | 14 (77%) | 0.09 |
| >2 | 7 (8%) | 4 (33%) | |
| *EGFR* **mutation—number (%)** | | | |
| Common mutations | 70 (82%) | 10 (56%) | **0.03** |
| Del19 | 45 (53%) | 7 (39%) | |
| L8585R | 25 (29%) | 3 (17%) | |
| Rare mutations | 4 (5%) | 1 (6%) | 1 |
| Insertion exon 20 | 5 (6%) | 2 (11%) | 0.60 |
| Complex mutations | 6 (7%) | 5 (28%) | **0.02** |
| EGFR copy gain | 11 (13%) | 1 (6%) | 0.69 |
| **Metastatic site—number (%)** | | | |
| Number of metastatic sites—mean | 2.64 | 2.94 | 0.33 |
| Liver | 23 (27%) | 2 (11%) | |
| Pericardium, pleural or peritoneum | 38 (45%) | 9 (50%) | |
| Bone | 46 (54%) | 10 (56%) | |
| Brain or meninges | 28 (33%) | 8 (44%) | |
| Adrenal glands | 11 (13%) | 5 (28%) | |
| Other | 58 (68%) | 2 (72%) | |
| **Co-mutations—number (%)** | | | |
| Any | 49 (58%) | 11 (61%) | 1 |
| TP53 | 41 (48%) | 10 (56%) | |
| CTNNB1 | 8 (9%) | 0 (0%) | |
| SMAD4 | 1 (1%) | 1 (6%) | |
| **Biological examination** | | | |
| LDH rate—average U/L | 227.4 | 360.7 | 0.11 |
| **First line of treatment** | | | |
| 3rd generation TKI | 26 (31%) | 7 (39%) | 0.58 |
| 2nd generation TKI | 12 (14%) | 3 (17%) | 1 |
| 1st generation TKI | 36 (42%) | 6 (33%) | 0.6 |
| Chemotherapy | 11 (12%) | 2 (12%) | 1 |

ECOG: eastern cooperative oncology group performance scale; EGFR: epidermal growth factor receptor; Del19: deletion exon 19; L858R: false sense insertion exon 21 L858R; Other: lung, soft tissue, lymph nodes SMAD4: Mothers against decapentaplegic homolog 4; CTNNB1: Cadherin-associated protein beta 1; LDH: lactate deshydrogénase. 3rd generation TKI: osimertinib, 2nd-generation TKI: afatinib, 1st generation TKI: gefinitib, erlotinib and one case erlotinib and ramucirumab

*n = 88 with available information on pack-years.

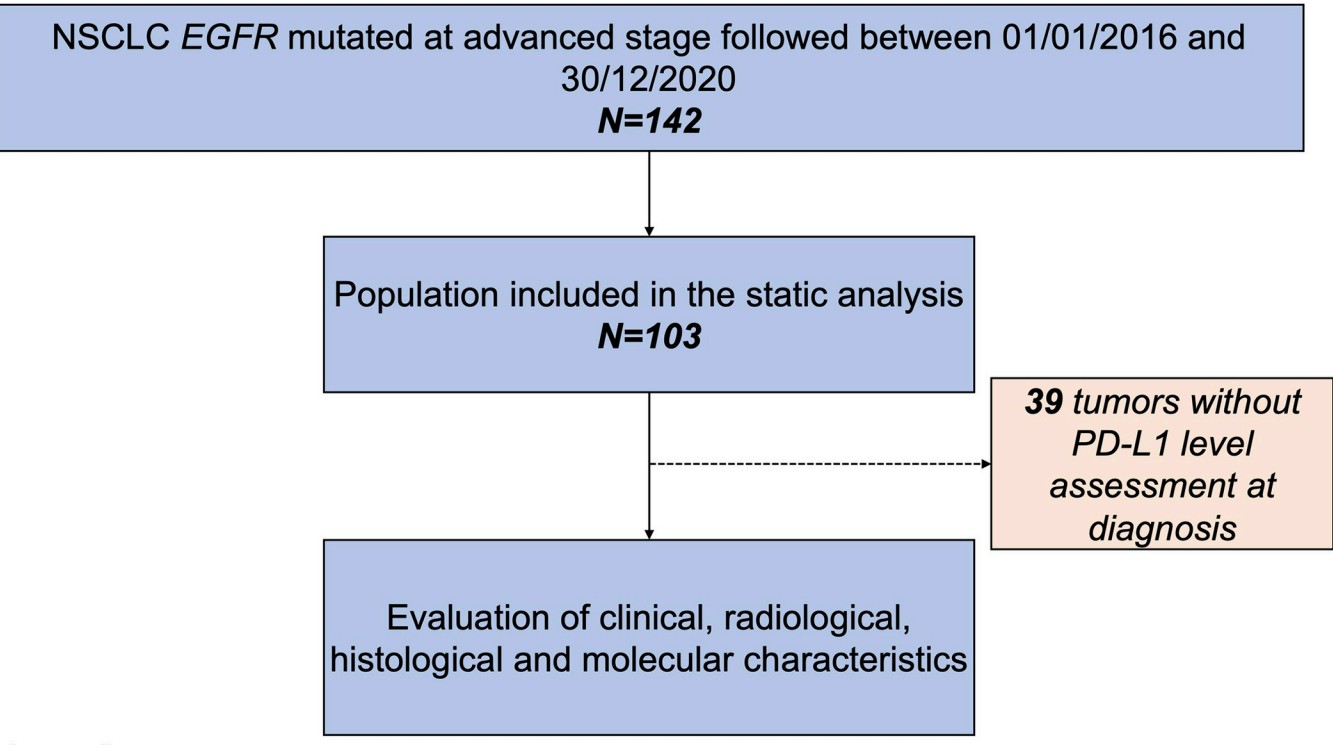

**Fig 1. Study flowchart based on inclusion and exclusion criteria.** The flowchart illustrates the process of patient selection for the study, considering both inclusion and non-inclusion criteria. Initially, a total of 103 patients were included in the study after excluding those who did not meet the specified criteria.

PD-L1 $\geq$ 50% group, although this difference did not reach statistical significance. The mean LDH level assessed at diagnosis tended to be higher in the PD-L1 $\geq$ 50% group, but this difference was not statistically significant. There was a non-significant trend towards a higher incidence of meningeal metastases in the PD-L1 $\geq$ 50% group (Table 2).

Regarding the *EGFR* mutation status, there were significantly more common mutations in the PD-L1 < 50% group (p = 0.03), while complex mutations were more prevalent in the PD-L1 $\geq$ 50% group (p = 0.02). There were no significant differences between the two groups in terms of rare mutations and exon 20 insertions. Co-mutations were observed in *TP53* (51/103 patients), *CTNNB1* (8/103 patients), *SMAD4* (2/103 patients), and *PIK3CA* (1/103 patients). The frequency of tumors with co-mutations did not differ significantly between the two groups. In the PD-L1 < 50% group, eight patients (9%) had a *CTNNB1* co-mutation, while none were observed in the PD-L1 $\geq$ 50% group (p = 0.34). There was no difference between the first-line treatments initiated between the two groups (Table 2).

### Response and survival according to PD-L1 level

Among patients with PD-L1 $\geq$ 50%, disease control was achieved after first-line treatment in 86% (12/14) of evaluable patients treated with *EGFR*-TKI and 50% (1/2) of those treated with chemotherapy, compared to 90% (64/71) and 81% (9/11) in the PD-L1 < 50% group, respectively. Two complete responses were observed in the PD-L1 < 50% group, while none were observed in the PD-L1 $\geq$ 50% group.

Survival curve analysis revealed no significant differences in progression-free survival (PFS) between the two groups (p = 0.38). Similarly, there were no significant differences in overall survival (OS) between the PD-L1 $\geq$ 50% and PD-L1 < 50% groups (p = 0.30) (Fig 2).

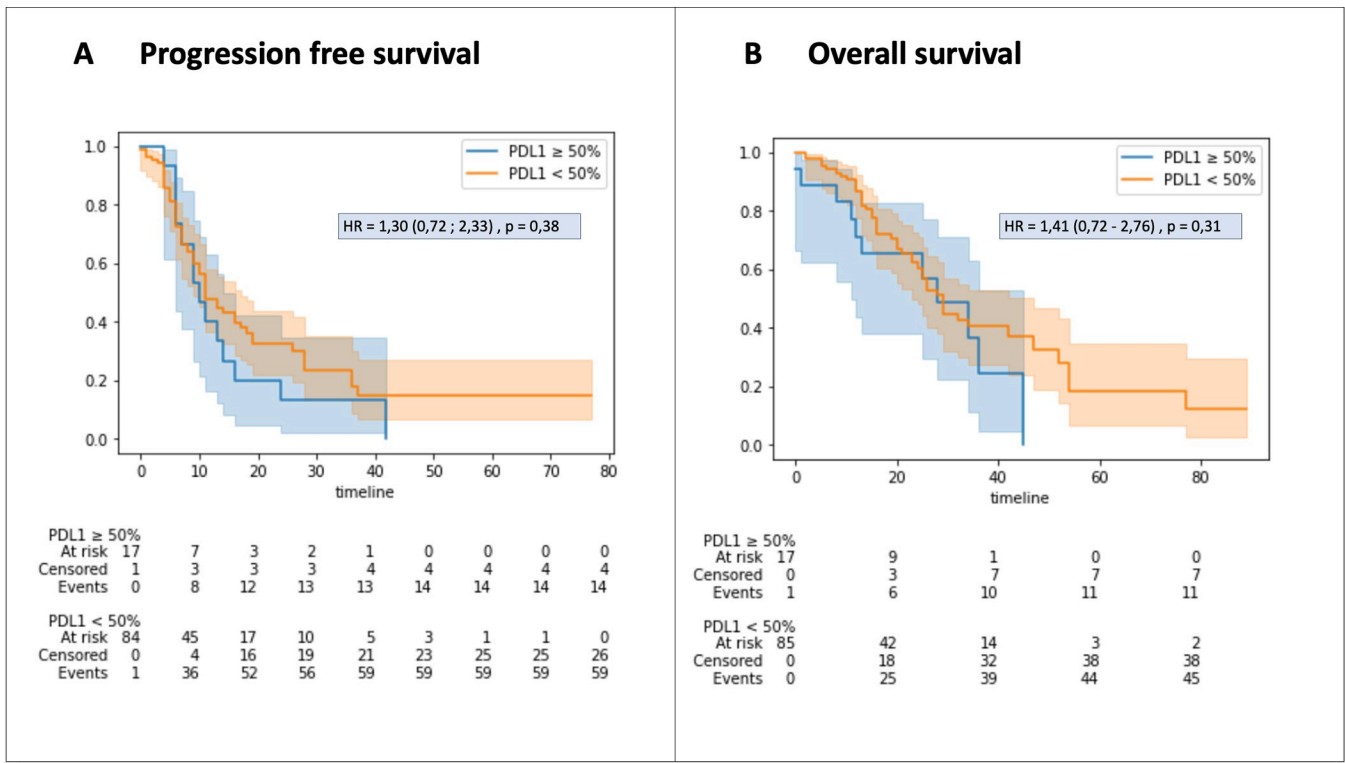

**Fig 2.** Progression-free survival (A) and overall survival (B) according to PD-L1 expression. Kaplan Meier curves are shown. High PD-L1 expression is defined as TPS ≥ 50%.

### Univariate and multivariate evaluation of survival

Univariate analysis demonstrated that liver and CNS metastases were significantly associated with decreased PFS and OS. Female gender was significantly associated with prolonged OS, while poor general condition at diagnosis (ECOG 3 and 4) and the presence of a complex *EGFR* mutation were associated with decreased OS. The presence of a *CTNNB1* co-mutation showed a non-significant association with longer PFS (Table 3). The univariate analysis of the 49 patients with available information on LDH levels did not show a significant association between high LDH levels and overall survival (HR 1.24, 95% CI 0.57–2.66; p = 0.59).

A multivariate analysis was performed, incorporating the variables of interest from the univariate analysis using the stepAIC method. Smoking, the presence of liver and brain metastases were significantly associated with shorter PFS. The presence of PD-L1 ≥ 50% was not significantly associated with shorter PFS (HR = 1.58; 95% CI [0.86–2.92]; p = 0.14) (Table 3). Male gender, age, poor general condition at treatment initiation, and the presence of liver and brain metastases were significantly associated with decreased OS. PD-L1 ≥ 50% was significantly associated with decreased OS (HR = 2.57; 95% CI [1.20–5.50]; p = 0.02). The occurrence of a common *EGFR* mutation showed a trend towards higher overall survival, although the association did not reach statistical significance (Table 3).

We conducted an exploratory analysis of 88 patients with documented pack-year histories (S1 Table) to investigate tobacco smoking's role in PD-L1 expression, EGFR genotype, and clinical outcomes. An increase in pack-years was significantly associated with reduced PFS in both univariate (HR 1.23, 95% CI 1.01–1.51; p = 0.04) and multivariate analysis (HR 1.38, 95% CI 1.06–1.79; p = 0.01), as well as diminished OS in univariate (HR 1.35, 95% CI 1.08–1.69; p = 0.008) and multivariate analysis (HR 1.50, 95% CI 1.12–1.95; p = 0.005) with increased smoking exposure.

**Table 3. Analysis of progression-free survival at first line of treatment and overall survival by patient characteristics.**

| PFS | | | | | OS | | | |
|---|---|---|---|---|---|---|---|---|
| | Univariate | | Multivariate | | Univariate | | Multivariate | |
| Characteristics | HR (95% CI) | p-value | HR (95% CI) | p-value | HR (95% CI) | p-value | HR (95% CI) | p-value |
| **Age** (n = 103) | 0.71 (0.20; 2.57) | 0.60 | | | 1.70 (0.45–6.44) | 0.44 | 4.93 (1.10–22.04) | **0.04** |
| **Gender** | | | | | | | | |
| Female (n = 68) | 0.80 (0.49–1.30) | 0.37 | | | 0.54 (0.32–0.92) | **0.02** | | |
| Male (n = 35) | 1.25 (0.77; 2.03) | | | | 1.86 (1.09–3.16) | | 2.77 (1.54–4.98) | **0.005** |
| **Smoking** | 1.28 (0.81–2.04) | 0.29 | 1.91 (1.13–3.26) | **0.02** | 1.29 (0.76–2.20) | 0.35 | | |
| Absent (n = 56) | | | | | | | | |
| Present (n = 47) | | | | | | | | |
| **PD-L1 ≥ 50%** (n = 18) | 1.30 (0.72; 2.33) | 0.38 | 1.58 (0.86–2.92) | 0.14 | 1.41 (0.72–2.76) | 0.31 | 2.57 (1.20–5.55) | **0.02** |
| **ECOG : 3–4** | 1.56 (0.61–3.82) | 0.37 | | | 7.22 (3.4–15.32) | **0.0001** | 10.69 (4.42–25.85) | **0.005** |
| **EGFR mutation** | | | | | | | | |
| Common (n = 80) | 1.08 (0.61–1.91) | 0.79 | | | 0.59 (0.32–1.09) | 0.09 | 0.60 (0.31–1.16) | 0.13 |
| Rare (n = 5) | 0.78 (0.24; 2.48) | 0.67 | | | 0.92 (0.29–2.95) | 0.89 | | |
| Complex (n = 11) | 1.19 (0.57–2.48) | 0.65 | | | 2.28 (1.07–4.88) | **0.03** | | |
| Ins exon 20 (n = 7) | 0.75 (0.27–2.06) | 0.57 | | | 1.02 (0.25–4.22) | 0.98 | | |
| **Co-mutation** | 0.87 (0.55–1.39) | 0.57 | 0.69 (0.43–1.12) | 0.14 | 1.02 (0.6–1.75) | 0.94 | | |
| TP53 (n = 51) | 1.05 (0.66–1.66) | 0.83 | | | 1.31 (0.77–2.22) | 0.32 | | |
| CTNNB1 (n = 8) | 0.54 (0.22; 1.36) | 0.19 | | | 0.44 (0.14–1.41) | 0.16 | | |
| **Metastasis** | | | | | | | | |
| Liver (n = 25) | 2.82 (1.66–4.78) | **0.0001** | 4.47 (2.40–8.35) | **0.01** | 2.74 (1.56–4.82) | **0.0001** | 5.80 (2.86–11.75) | **0.01** |
| CNS (n = 33) | 2.27 (1.39–3.71) | **0.0001** | 2.56 (1.54–4.27) | **0.01** | 1.84 (1.07–3.16) | **0.03** | 1.99 (1.13–3.52) | 0.02 |
| Bone (n = 56) | 1.45 (0.91–2.32) | 0.12 | | | 1.17 (0.69–1.99) | 0.56 | | |

CNS: Central nervous system; ECOG: eastern cooperative oncology group performance scale

Variables significantly associated with PFS in univariate analysis included ECOG score > 3 or 4, and presence of hepatic, osseous, or brain metastases. Multivariate analysis identified PD-L1 expression level, ECOG status, and presence of hepatic and brain metastases as significant factors. For OS, significant univariate variables included gender, ECOG status, presence of complex EGFR mutation, and hepatic and brain metastases. Multivariate analysis showed associations with PD-L1 expression level, ECOG status, presence of common EGFR mutation, and presence of hepatic metastases (see S1 Table).

## Progression after the first line of treatment

In the PD-L1 < 50% group, 68% (58/85) of patients experienced disease progression after first-line treatment, compared to 72% (13/18) in the PD-L1 ≥ 50% group. Brain progression occurred in 22% (19/85) of patients in the PD-L1 < 50% group and 27% (5/18) in the PD-L1 ≥ 50% group. The rate of *de novo* resistance, defined as progression within 3 months, was 7% (6/85) in the PD-L1 < 50% group and 17% (3/18) in the PD-L1 ≥ 50% group (p = 0.19). Excluding patients treated with 3rd generation TKIs and chemotherapy, 42% (16/38) of patients in the PD-L1 < 50% group and 57% (4/7) in the PD-L1 ≥ 50% group had a T790M resistance mutation. Among all patients treated with TKIs, in the PD-L1 < 50% group, progression was associated with 2 *EGFR* C797S resistance mutations, 2 *MET* amplifications, one *ERBB2* amplification, and one *MET* exon 14 mutation. In the PD-L1 ≥ 50% group, one case of *ERBB2* amplification and one case of *BRAF* V600E mutation were observed.

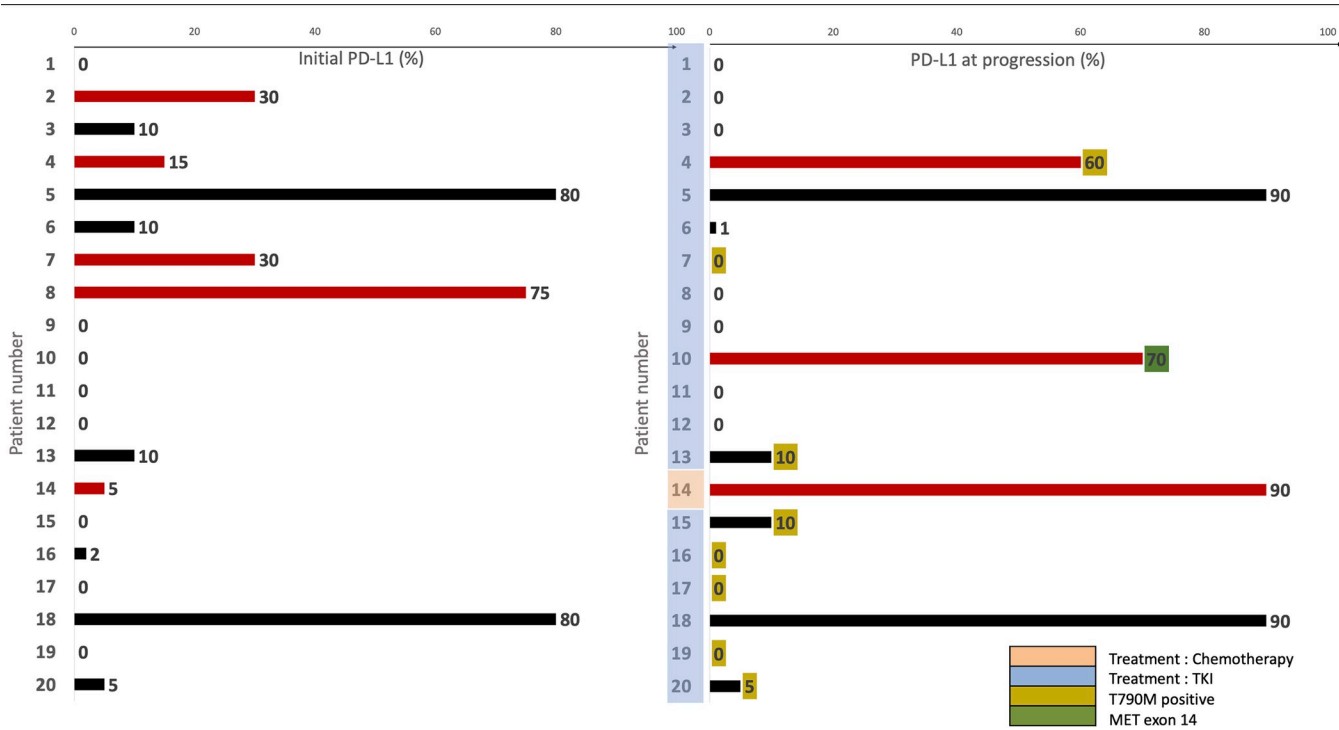

**Fig 3. Changes in PD-L1 levels before treatment and at progression after first-line therapy.** The figure illustrates the evaluation of PD-L1 levels in twenty patients (numbered from 0 to 20) before treatment (on the left) and the subsequent changes in PD-L1 levels after first-line therapy (on the right). Patients with a significant change (>30%) in PD-L1 levels are highlighted in red.

During the second line of treatment, in the PD-L1 < 50% group, 18 (32%) patients received chemotherapy, 24 (42%) received osimertinib, 10 (17%) received a first- or second-generation TKI, 1 received immunotherapy, 1 received tepotinib, 1 received amivantamab, and 1 received a combination of gefitinib and vemurafenib. In the PD-L1 ≥ 50% group, 2 (15%) patients received chemotherapy, 5 (38%) received osimertinib treatment, 4 (31%) received first- or second-generation TKI treatment, and 2 (15%) received immunotherapy.

## Assessment of PD-L1 levels at progression after first-line treatment

PD-L1 levels at progression after the first line of treatment were available for 20 patients. Notably, 30% (6/20) of tumors exhibited a significant change (i.e., a TPS modification > 20%) in PD-L1 levels after the first line of treatment. Among these cases, three showed an increase from <50% to ≥ 50% PD-L1 TPS. One of these cases had a MET exon 14 resistance mutation, and another had a T790M mutation. In three other cases, PD-L1 expression decreased after *EGFR*-TKI treatment, including one case with T790M mutation (Fig 3).

All patients except for patient 14 received TKI therapy, while patient 14 underwent chemotherapy. Among the patients, 4, 7, 13, 15, 16, 17, 19, and 20 exhibited a T790M mutation at progression, indicative of resistance to 1st and 2d generation EGFR-TKIs. Additionally, patient 10 showed a *MET* exon 14 mutation at progression.

## Discussion

We evaluated the prognostic significance of PD-L1 expression in metastatic *EGFR*-mutated NSCLC patients, categorizing them into PD-L1 < 50% and PD-L1 ≥ 50% groups. Our

findings indicate that patients with PD-L1 ≥ 50% had distinct clinical, biological, and molecular characteristics compared to those with PD-L1 < 50%. Specifically, the PD-L1 ≥ 50% group showed a tendency towards higher smoking rates, higher blood LDH levels, more complex *EGFR* mutations, lower prevalence of common *EGFR* mutations, and an absence of *CTNNB1* co-mutation. Furthermore, we observed a higher frequency of *de novo* resistance and a significant difference in OS in multivariate analysis, suggesting that high PD-L1 expression may be associated with a worse prognosis in this patient population.

Consistent with our findings, previous studies have also suggested that PD-L1 expression can serve as a prognostic factor in *EGFR*-mutated NSCLC. Liu et al., in an Australian multi-center study, demonstrated that PD-L1 levels ≥ 50% were associated with shorter PFS and OS. They reported a higher frequency of *de novo* resistance in the PD-L1 ≥ 50% group [13, 17]. Similarly, a meta-analysis of 991 patients showed that PD-L1 expression ≥ 50% was significantly associated with shorter PFS [18]. Although our study did not observe a significant difference in PFS or OS in univariate analysis, the retrospective nature and small sample size may have limited the statistical power. However, the significant difference in OS in multivariate analysis suggests that PD-L1 expression may still be an important prognostic factor, considering potential confounding factors such as complex *EGFR* mutations associated with higher PD-L1 expression and a more uncertain response to EGFR TKIs or liver metastasis [19]. Given the retrospective nature of this study, the analysis of the side effects of the different treatments received, according to PD-L1 expression, could not be cataloged and analyzed in this article, which represents a potential bias to consider when interpreting these result.

The mechanisms regulating PD-L1 expression in *EGFR*-mutated tumors remain unclear as they depend both on endogenous signaling linked to genomic alterations, but also on exogenous inflammatory signals and induction via the interferon pathway [20]. *In vitro*, the expression of mutated *EGFR* in immortalized bronchial epithelial cells induces an increase in PD-L1 levels, in contrast to the expression of KRAS G12V [19]. Nevertheless, in human tissues, PD-L1 levels are generally lower in *EGFR*-mutated NSCLC compared to wild-type *EGFR* NSCLC [9]. This may be due to the absence of immune signaling in these tumors. Indeed, the mutational burden, defined by the number of mutations per megabase in the coding regions of the tumor genome, is significantly lower in *EGFR*-mutated NSCLC compared to wild-type *EGFR* tumors [21] and it has been shown that *EGFR* mutations are associated with an immunosuppressive tumor microenvironment [22–24] with a lower CD8+ T cell infiltration than in wild type *EGFR* NSCLC. *CTNNB1* co-mutation were only observed in the group with low PD-L1 expression. *CTNNB1* co-mutation has been described to be more frequent in *EGFR*-mutated NSCLC than in wild-type *EGFR* [25] and its absence has been associated with a poor prognosis in *EGFR*-mutated NSCLC [26]. *CTNNB*1 mutation leads to the activation of the Wnt pathway [27], and prevents the establishment of an inflammatory tumor microenvironment by T cells in various types of cancer [28]. Thus, a *CTNNB1* co-mutation would reinforce the immunosuppressive microenvironment. Tumor infiltration by CD8 T cells was not assessed in our study, which could have been of interest as it has been described to predict survival after immune checkpoint inhibitor treatment in *EGFR*-mutated NSCLC [29].

A higher PD-L1 expression after TKI treatment has been reported in the literature, especially for those without a T790M mutation [30]. This observation suggests that some TKI resistance mechanism may induce PD-L1 expression. We observed a significant increase in PD-L1 expression at progression after first line TKI treatment in only 2 of 19 cases, one having a *MET exon 14* mutation and the other with a T790M mutation, and in 3 cases, the expression of PD-L1 appeared to be lower at relapse. It should be noted that PD-L1 evaluation by immunohistochemistry was done using different antibodies that may introduce some difficulties in the interpretation [31]. The 3 antibodies used in this study were reported to have good

concordance [32], although their results may be affected by pre-analytical factors such as decalcification and fixation conditions [33]. In addition, inter-tumor agreement (between the primary tumor and the secondary lesion) and intra-tumor agreement (between different areas of the tumor) may vary due to tumor heterogeneity, especially in the case of PD-L1 values between 1% and 50% [34].

## Conclusion

High PD-L1 expression was associated with uncommon and complex EGFR mutations, as well as a higher frequency of *de novo* resistance and independently predicted poor survival. Further investigations with larger prospective cohorts are warranted to validate and expand upon our findings, as well as to unravel the complex interplay between *EGFR* mutations, immune signaling, and PD-L1 expression in *EGFR*-mutated NSCLC.

## Supporting information

**S1 File. Basic demographic and clinical data.**
(XLSX)

**S1 Table. Analysis of progression-free survival and overall survival in the 88 patients for whom smoking pack-years information was available.**
(DOCX)

## Author Contributions

**Conceptualization:** Hélène Blons, Elizabeth Fabre, Bastien Rance, Karen Leroy, Marie Wislez.

**Data curation:** Jeremy Slomka, Hugo Berthou.

**Formal analysis:** Jeremy Slomka.

**Investigation:** Jeremy Slomka, Elizabeth Fabre, Gary Birsen, Jennifer Arrondeau, Karen Leroy, Marie Wislez.

**Methodology:** Elizabeth Fabre, Ivan Lerner, Karen Leroy, Marie Wislez.

**Resources:** Audrey Mansuet-Lupo, Hélène Blons, Marco Alifano, Jeanne Chapron, Gary Birsen, Laure Gibault, Jennifer Arrondeau.

**Software:** Bastien Rance.

**Supervision:** Karen Leroy, Marie Wislez.

**Validation:** Ivan Lerner.

**Writing – original draft:** Jeremy Slomka.

**Writing – review & editing:** Karen Leroy, Marie Wislez.

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
