## [Decision Letter · Decision Letter 0]

3 Apr 2024

PONE-D-24-01682Clinical and Molecular Characteristics Associated with High PD-L1 expression in EGFR-Mutated Lung AdenocarcinomaPLOS ONE

Dear Dr. Wislez,

Thank you for submitting your manuscript to PLOS ONE. After careful consideration, we feel that it has merit but does not fully meet PLOS ONE’s publication criteria as it currently stands. Therefore, we invite you to submit a revised version of the manuscript that addresses the points raised during the review process.

**The reviewers have recommended publication, but also suggest significant revisions to your manuscript.  Therefore, I invite you to respond to the reviewers' comments and revise your manuscript.**

We look forward to receiving your revised manuscript.

Kind regards,

Fumihiro Yamaguchi

Academic Editor

PLOS ONE

Journal Requirements:

2. In the online submission form, you indicated that All the data from this study have been imported into a REDCAP database, which cannot be shared publicly as it is protected by our institution. However, at your request, this data can be extracted and sent to you.

Additional Editor Comments:

This is a well-designed study, but it needs significant revision.

Reviewers' comments:

Reviewer's Responses to Questions

**Comments to the Author**

1. Is the manuscript technically sound, and do the data support the conclusions?

Reviewer #1: Partly

Reviewer #2: Yes

Reviewer #3: Yes

Reviewer #4: Yes

2. Has the statistical analysis been performed appropriately and rigorously? 

Reviewer #1: I Don't Know

Reviewer #2: Yes

Reviewer #3: Yes

Reviewer #4: Yes

3. Have the authors made all data underlying the findings in their manuscript fully available?

Reviewer #1: Yes

Reviewer #2: Yes

Reviewer #3: Yes

Reviewer #4: Yes

4. Is the manuscript presented in an intelligible fashion and written in standard English?

Reviewer #1: Yes

Reviewer #2: Yes

Reviewer #3: Yes

Reviewer #4: Yes

5. Review Comments to the Author

**Reviewer #1**: Fascinating article. Would really appreciate a breakdown as to which TKIs patients were treated with though. I'm not comfortable supporting some of the conclusions (ex: rates of CNS relapse, OS) without knowing how many patients were treated with osimertinib vs something like gefitinib which has significantly worse CNS penetration and OS. A discussion as to the biology of EGFR and the mechanism for why PD-L1 can be upregulated would be interesting and useful here (for example EGFR signaling may induce PD-L1 expression in bronchial cells).

**Reviewer #2**: In this manuscript, the authors analyzed the prognostic significance of PD-L1 expression in a retrospective cohort of patients with metastatic EGFR-mutated NSCLC. Although the authors provided important data and suggestions for clinical practice, these data have already been published in other papers.

Comments:

1) The authors show that high PD-L1 expression (≥50% TPS) is significantly associated with decreased OS and not significantly associated with shorter PFS in multivariate analysis. Please provide the information of second-line therapy such as chemotherapy, immunotherapy, or the other EGFR-TKIs in each PD-L1 < 50% and PD-L1 ≥ 50% groups.

2) The authors show a higher frequency of de novo resistance in the PD-L1 ≥ 50% group compared with PD-L1 < 50% group. Please provide the information of first-line therapy such as 1st, 2nd or 3rd generation EGFR-TKI in each PD-L1 < 50% and PD-L1 ≥ 50% groups.

3) If possible, please provide the toxicity profiles of the first-line EGFR-TKI treatment in each PD-L1 < 50% and PD-L1 ≥ 50% groups.

**Reviewer #3**: This is a retrospective observational study that aims to explore the relationship between the level of PDL1 expression and the paterns of EGFR mutation, as well as outcomes of the treatment. Although the small sample size makes it difficult to draw the compellent conclusions, the current study had been designed strictly, conducted well, and analyzed properly. Some findings as high PDL1 expression is associated with complex EGFR mutations may give the resonable cause that PDL1 over expression leads to the poorer outcome in patients with EGFR mutation. The shortcomings of the study had been discussed deeply and adequately. The above points make this article readable and could give some enlightenment to the readers. I have just minor comments for the authors.

1. Materials and methods-Assessment of PD-L1 level and molecular characterization: “EGFR mutation and co-mutation status (specifically TP53, CTNNB1, SMAD4, or 129 PIK3CA) weas identified during routine care using next-generation sequencing.”. I don’t understand the word “WEAS”, should it be “were” ?

2. S1 table can not be displayed. Since the number of complex mutation is not big, I suggest describing them in the RESULTS part.

3. TABLE1-SMOKING STATUS: “ABSENT” means never smoke? Or quit smoking?

4. Conclusions in the tail of the text is too complex, it is better putting it at the end of DISCUSSION part. I suggest use the conclusions similar with that in the ABSTRACT.

**Reviewer #4**: Slomka et al. investigated the association between high PD-L1 expression and overall survival (OS) in a retrospective EGFR-mutant lung adenocarcinoma cohort. They observed that high PD-L1 was associated with complex EGFR mutations, poor clinical characteristics (liver and brain metastasis, poor performance), primary resistance, and shorter OS. The manuscript is well written, and the analyses are well conducted; however, it does not add new information compared to previous reports investigating the same question.

Major comments

The authors need to make the novelty of their work apparent. Various previous works have investigated the association between PD-L1 expression and survival in EGFR-mutant NSCLC and, in general, demonstrated that PD-L1 is associated with worse PFS or OS (Tang et al. Oncotarget. 2025; Yang et al. Eur J cancer. 2020; Peng et al. World J Surgery Oncol. 2021; Shiozawa et al. Anticancer Res. 2022; Ji et al. Cancer Bill Therapy. 2016; Bai et al. Cancer Biol Med. 2018; Liu et al. Lung Cancer. 2021) or primary resistance (Lan et al. Medicine (Baltimore). 2021).

The authors state they used Fisher's exact test to test the association between PD-L1 expression and other categorical variables; nevertheless, they show various p-values in Table 1 when there should be only one p-value for each variable. How do they explain that? Besides, some variables have too many categories. I recommend collapsing some of them to reduce dimensionality.

There are some hints that the worse prognosis associated with high PD-L1 expression can reflect its association with tumors that are genetically more complex, probably due to smoking exposure. It would be interesting to test if there is an association (Wilcoxon's test) or correlation (Spearman's correlation test) between smoking load (pack-years) and PD-L1 expression and type of EGFR mutation. Besides, the authors included smoking in univariate and multivariate only as a categorical variable. I recommend testing its impact on OS and PFS as a continuous variable (pack-years).

The impact of PD-L1 on OS might reflect different treatment patterns. The authors did not describe the treatment given to the patients (1st Gen TKI and 2nd Gen TKI x 3rd Gen TKI x QT) and its association with OS.

Were the tumor samples submitted to comprehensive genomic analysis? If that is the case, it would be interesting to describe the frequency of co-mutations, investigate their impact and of TMB on OS, and their association with the EGFR mutation type and PD-L1 expression.

Minor comments

The authors declare they used different antibody clones to test PD-L1 expression, but they do not show what clones were employed or how many samples were tested with each one.

6. PLOS authors have the option to publish the peer review history of their article (what does this mean?). If published, this will include your full peer review and any attached files.

Reviewer #1: **Yes: **Whitney Elizabeth Lewis

Reviewer #2: No

Reviewer #3: No

Reviewer #4: No

---

## [Author Response · Author response to Decision Letter 0]

16 May 2024

Dear Dr Yamagushi,

Thank you for considering our manuscript titled "Clinical and Molecular Features Associated with High PD-L1 Expression in EGFR-Mutated Metastatic Lung Adenocarcinoma" (PONE-D-24-01682). We have diligently revised the text and tables in response to the reviewers' comments (detailed responses provided below). We believe these revisions have significantly enhanced the quality of our manuscript.

We have ensured that our manuscript and figure files comply with PLOS ONE's style guidelines. Additionally, we have updated the ORCID iD of the corresponding author and included anonymized study data as a supplementary file, aligning with the journal's request to facilitate access for fellow researchers.

Gary Birsen had participated to this study, as was mentioned in the author contribution section, and was unfortunately removed from the list of co-authors in the first submission. The list of authors has been corrected in the revised version.

We are confident that this revised manuscript meets the standards for publication in PLOS ONE. Thank you for considering our work.

Yours sincerely,

Professor Marie Wislez, on behalf the coauthors

Corresponding author: 

Professor Marie Wislez Pneumology, Cochin Hospital Paris, Université Paris Cité, Faculty of Medicine, Paris, France; 

telephone: 33 158413091 

e-mail: marie.wislez@aphp.fr

Subject: Reponse to Reviewers - "Clinical and molecular features associated with high PD-L1 expression in EGFR-mutated metastatic lung adenocarcinoma"

Reviewer #1: Fascinating article. Would really appreciate a breakdown as to which TKIs patients were treated with though. I'm not comfortable supporting some of the conclusions (ex: rates of CNS relapse, OS) without knowing how many patients were treated with osimertinib vs something like gefitinib which has significantly worse CNS penetration and OS. 

Thank you for your thoughtful and constructive feedback. We acknowledge the importance of detailing the treatments received in the first line of therapy to enhance the clarity and comprehension of our manuscript. As per your suggestion, we have incorporated a comprehensive breakdown of the different treatments administered to patients during the first line into Table 2 (formerly Table 1). Additionally, in the paragraph "Progression after the First Line of Treatment" (page 14, lines 270-275), we have provided a description of the various treatments utilized in the second line.

Upon analysis, we did not identify significant differences in the type of first-line treatment received between the two groups. Nonetheless, we recognize the need for caution in interpreting these findings due to the limitations posed by our relatively small sample sizes.

A discussion as to the biology of EGFR and the mechanism for why PD-L1 can be upregulated would be interesting and useful here (for example EGFR signaling may induce PD-L1 expression in bronchial cells).

As noted by the reviewer, we discussed in the discussion section (page 16, lines 316-328) the complexity of PD-L1 regulation, which is influenced by both endogenous EGFR signaling and exogenous signals such as interferon that are associated with the activation of anti-tumor immunity. To support this point, we have incorporated a specific reference in the field (Sun C, Mezzadra R, Schumacher TN. Regulation and Function of the PD-L1 Checkpoint. Immunity. 2018 Mar 20;48(3):434-452. doi: 10.1016/j.immuni.2018.03.014. PMID: 29562194; PMCID: PMC7116507).

The observed variation in PD-L1 expression among EGFR-mutated tumors may indeed be attributed to differences in anti-tumor immunity and immune cell infiltration. However, it is important to note that we were unable to assess this aspect in our retrospective study.

Reviewer #2: In this manuscript, the authors analyzed the prognostic significance of PD-L1 expression in a retrospective cohort of patients with metastatic EGFR-mutated NSCLC. Although the authors provided important data and suggestions for clinical practice, these data have already been published in other papers.

Comments:

1) The authors show that high PD-L1 expression (≥50% TPS) is significantly associated with decreased OS and not significantly associated with shorter PFS in multivariate analysis. Please provide the information of second-line therapy such as chemotherapy, immunotherapy, or the other EGFR-TKIs in each PD-L1 < 50% and PD-L1 ≥ 50% groups.

Thank you for this very interesting remark; second-line treatments could be a relevant avenue to explain the association of PD-L1 with OS, and absence of difference in PFS. Therefore, we have added in the paragraph 'Progression after the First Line of Treatment' on page 14, lines 270-275 the description of the different treatments used in the second line. There was not obvious difference that could explain our results.

2) The authors show a higher frequency of de novo resistance in the PD-L1 ≥ 50% group compared with PD-L1 < 50% group. Please provide the information of first-line therapy such as 1st, 2nd or 3rd generation EGFR-TKI in each PD-L1 < 50% and PD-L1 ≥ 50% groups. 

The treatments administered in the first line of therapy are indeed crucial for a comprehensive understanding of our manuscript. To address this, we have incorporated a detailed breakdown of the different treatments administered to patients during the first line in Table 2 (previously Table 1). Notably, our analysis did not reveal any significant difference in the type of first-line treatment received between the two groups. However, we recognize the importance of interpreting these findings cautiously given the limitations posed by our relatively small sample sizes.

3) If possible, please provide the toxicity profiles of the first-line EGFR-TKI treatment in each PD-L1 < 50% and PD-L1 ≥ 50% groups.

Regarding the toxicities of the treatments, it would have been very pertinent to describe the side effects. However, due to the retrospective nature of this study involving routine care data, it was not possible to obtain this information reliably; therefore, this data was not collected.

Reviewer #3: This is a retrospective observational study that aims to explore the relationship between the level of PDL1 expression and the paterns of EGFR mutation, as well as outcomes of the treatment. Although the small sample size makes it difficult to draw the compellent conclusions, the current study had been designed strictly, conducted well, and analyzed properly. Some findings as high PDL1 expression is associated with complex EGFR mutations may give the resonable cause that PDL1 over expression leads to the poorer outcome in patients with EGFR mutation. The shortcomings of the study had been discussed deeply and adequately. The above points make this article readable and could give some enlightenment to the readers. I have just minor comments for the authors.

1. Materials and methods-Assessment of PD-L1 level and molecular characterization: “EGFR mutation and co-mutation status (specifically TP53, CTNNB1, SMAD4, or 129 PIK3CA) weas identified during routine care using next-generation sequencing.”. I don’t understand the word “WEAS”, should it be “were”?

Thank you for these very relevant remarks. Regarding the sentence about the changes, the term "WEAS" is indeed a typographical error and should have been "were." This modification has been made on page 6 line 129.

2. S1 table can not be displayed. Since the number of complex mutation is not big, I suggest describing them in the RESULTS part.

As requested by the reviewer, the complex mutations have been displayed in a specific table. Revised manuscript: Table 1, page 3, line 140. The former Table 1 becomes Table 2 and Table 2 becomes Table 3, respectively.

3. TABLE1-SMOKING STATUS: “ABSENT” means never smoke? Or quit smoking?

For clarity and as suggested, we have replaced smoking status in Table 2 (former Table 1) with "never smoked" or "current/former smoker."

4. Conclusions in the tail of the text is too complex, it is better putting it at the end of DISCUSSION part. I suggest use the conclusions similar with that in the ABSTRACT.

As requested by the reviewer, we have simplified the conclusions as follows:

“High PD-L1 expression was associated with uncommon and complex EGFR mutations, as well as a higher frequency of de novo resistance and independently predicted poor survival. Further investigations with larger prospective cohorts are warranted to validate and expand upon our findings, as well as to unravel the complex interplay between EGFR mutations, immune signaling, and PD-L1 expression in EGFR-mutated NSCLC”.

Reviewer #4: Slomka et al. investigated the association between high PD-L1 expression and overall survival (OS) in a retrospective EGFR-mutant lung adenocarcinoma cohort. They observed that high PD-L1 was associated with complex EGFR mutations, poor clinical characteristics (liver and brain metastasis, poor performance), primary resistance, and shorter OS. The manuscript is well written, and the analyses are well conducted; however, it does not add new information compared to previous reports investigating the same question.

Major comments

The authors need to make the novelty of their work apparent. Various previous works have investigated the association between PD-L1 expression and survival in EGFR-mutant NSCLC and, in general, demonstrated that PD-L1 is associated with worse PFS or OS (Tang et al. Oncotarget. 2025; Yang et al. Eur J cancer. 2020; Peng et al. World J Surgery Oncol. 2021; Shiozawa et al. Anticancer Res. 2022; Ji et al. Cancer Bill Therapy. 2016; Bai et al. Cancer Biol Med. 2018; Liu et al. Lung Cancer. 2021) or primary resistance (Lan et al. Medicine (Baltimore). 2021).

Thank you for your valuable feedback. The sources you provided are indeed highly relevant to our study. We have incorporated these references into the article as references 10-15, 17 and 18, specifically on page 4, line 83 and page 15, line 308.

The authors state they used Fisher's exact test to test the association between PD-L1 expression and other categorical variables; nevertheless, they show various p-values in Table 1 when there should be only one p-value for each variable. How do they explain that? Besides, some variables have too many categories. I recommend collapsing some of them to reduce dimensionality.

We conducted Fisher's tests on the categorical variables, presenting a p-value for each. For the continuous variables, we included p-values from the Mann-Whitney test. To enhance clarity, we have revised the Materials and Methods section, specifically on page 7, line 149. Notably, we reorganized the categories, such as performance status below or above 2, and combined certain metastatic sites in Table 2 (formerly Table 1). Statistical tests were performed for each type of EGFR mutation, which was critical for addressing the study's key question.

There are some hints that the worse prognosis associated with high PD-L1 expression can reflect its association with tumors that are genetically more complex, probably due to smoking exposure. It would be interesting to test if there is an association (Wilcoxon's test) or correlation (Spearman's correlation test) between smoking load (pack-years) and PD-L1 expression and type of EGFR mutation. 

This observation is indeed pertinent. We investigated the relationship between pack-years and EGFR common mutations (p=0.2) as well as EGFR complex mutations (p=0.5) among the 88 patients for whom pack-year data was available using a Wilcoxon test. Our analysis did not reveal any significant differences in these associations.

With respect to PD-L1 expression, no significant difference was observed based on smoking status, and a similar trend was observed for the average number of pack-years (p=0.1). It's worth noting that these subgroups consisted of a limited number of patients, which may have contributed to the lack of observed differences. 

Besides, the authors included smoking in univariate and multivariate only as a categorical variable. I recommend testing its impact on OS and PFS as a continuous variable (pack-years).

We encountered 25 missing data points regarding the amount of tobacco smoked by patients in the study. As suggested, we performed both univariate and multivariate analyses among the 88 patients for whom we had complete data. We observed a correlation between progression-free survival (PFS) and overall survival (OS) and the number of pack-years. These findings have been incorporated into the section titled "Univariate and Multivariate Evaluation of Survival" on page 13, lines 246-255. Additionally, the table summarizing the univariate and multivariate analyses for this group of 88 patients has been included in the supplementary data (S1 Table).

The impact of PD-L1 on OS might reflect different treatment patterns. The authors did not describe the treatment given to the patients (1st Gen TKI and 2nd Gen TKI x 3rd Gen TKI x QT) and its association with OS.

The treatments administered in the first line of therapy are crucial for a comprehensive understanding of the manuscript. To address this, we have included a detailed summary of the different treatments administered to patients during the first line in Table 2 (previously Table 1). Despite our analysis not revealing any significant differences in the type of first-line treatment received between the two groups, it is important to interpret these results cautiously due to the relatively small sample sizes.

Were the tumor samples submitted to comprehensive genomic analysis? If that is the case, it would be interesting to describe the frequency of co-mutations, investigate their impact and of TMB on OS, and their association with the EGFR mutation type and PD-L1 expression.

As described in the methods section (page 6, line 128), tumor samples were analyzed using a next-generation sequencing (NGS) panel targeting hotspot regions of 22 genes. Recurrent co-mutations identified in this analysis included mutations in TP53, CTNNB1, and SMAD4, with one case exhibiting a BRAF exon 11 co-mutation. However, it's important to note that this panel does not allow the evaluation of TMB.

Minor comments

The authors declare they used different antibody clones to test PD-L1 expression, but they do not show what clones were employed or how many samples were tested with each one.

Regarding the antibodies used for PD-L1 evaluation, we have modified the text to make it more explicit. Specifically, the E1L3N antibody was used in 38 cases, the 22C3 antibody in 31 cases, and the QR1 antibody in 34 cases.

---

## [Decision Letter · Decision Letter 1]

9 Jun 2024

PONE-D-24-01682R1Clinical and Molecular Characteristics Associated with High PD-L1 expression in EGFR-Mutated Lung AdenocarcinomaPLOS ONE

Dear Dr. Wislez,

Thank you for submitting your manuscript to PLOS ONE. After careful consideration, we feel that it has merit but does not fully meet PLOS ONE’s publication criteria as it currently stands. Therefore, we invite you to submit a revised version of the manuscript that addresses the points raised during the review process.

**ACADEMIC EDITOR: **The reviewers have recommended publication, but also suggest some minor revisions to your manuscript.  Therefore, I invite you to respond to the reviewers' comments and revise your manuscript.

We look forward to receiving your revised manuscript.

Kind regards,

Fumihiro Yamaguchi

Academic Editor

PLOS ONE

Journal Requirements:

Reviewers' comments:

Reviewer's Responses to Questions

**Comments to the Author**

1. If the authors have adequately addressed your comments raised in a previous round of review and you feel that this manuscript is now acceptable for publication, you may indicate that here to bypass the “Comments to the Author” section, enter your conflict of interest statement in the “Confidential to Editor” section, and submit your "Accept" recommendation.

Reviewer #2: (No Response)

Reviewer #3: All comments have been addressed

Reviewer #4: All comments have been addressed

2. Is the manuscript technically sound, and do the data support the conclusions?

Reviewer #2: (No Response)

Reviewer #3: Yes

Reviewer #4: Yes

3. Has the statistical analysis been performed appropriately and rigorously? 

Reviewer #2: (No Response)

Reviewer #3: Yes

Reviewer #4: Yes

4. Have the authors made all data underlying the findings in their manuscript fully available?

Reviewer #2: (No Response)

Reviewer #3: Yes

Reviewer #4: Yes

5. Is the manuscript presented in an intelligible fashion and written in standard English?

Reviewer #2: (No Response)

Reviewer #3: Yes

Reviewer #4: Yes

6. Review Comments to the Author

Reviewer #2: The authors have responded to my review comments by adding some analytical results, demonstrating that they do not affect the original conclusions of the authors. However, toxicities by EGFR-TKI according to PD-L1 expressions had not been presented in the revised manuscript because they do not have representative data. This should be noted in the discussion as a limitation.

Reviewer #3: my comments had been addressed well by the authors.

Reviewer #4: The points raised by all the reviewers were adequately addressed by the authors. I have no further comments.

7. PLOS authors have the option to publish the peer review history of their article (what does this mean?). If published, this will include your full peer review and any attached files.

Reviewer #2: No

Reviewer #3: **Yes: **Jiang Zhu

Reviewer #4: No

---

## [Author Response · Author response to Decision Letter 1]

17 Jun 2024

Reviewer #2: The authors have responded to my review comments by adding some analytical results, demonstrating that they do not affect the original conclusions of the authors. However, toxicities by EGFR-TKI according to PD-L1 expressions had not been presented in the revised manuscript because they do not have representative data. This should be noted in the discussion as a limitation.

Thank you for your feedback. Indeed, this is a limitation that we did not address in the text. We have therefore added the following sentence: "Given the retrospective nature of this study, the analysis of the side effects of the different treatments received, according to PD-L1 expression, could not be cataloged and analyzed in this article, which represents a potential bias to consider when interpreting these results" on page 15, line 315-318.

Reviewer #3: my comments had been addressed well by the authors.

We thank you for your review and your various comments.

Reviewer #4: The points raised by all the reviewers were adequately addressed by the authors. I have no further comments.

We thank you for your review and your various comments.

---

## [Decision Letter · Decision Letter 2]

2 Jul 2024

Clinical and Molecular Characteristics Associated with High PD-L1 expression in EGFR-Mutated Lung Adenocarcinoma

PONE-D-24-01682R2

Dear Dr. Wislez,

We’re pleased to inform you that your manuscript has been judged scientifically suitable for publication and will be formally accepted for publication once it meets all outstanding technical requirements.

Kind regards,

Fumihiro Yamaguchi

Academic Editor

PLOS ONE

Additional Editor Comments (optional):

Reviewers' comments:

Reviewer's Responses to Questions

**Comments to the Author**

1. If the authors have adequately addressed your comments raised in a previous round of review and you feel that this manuscript is now acceptable for publication, you may indicate that here to bypass the “Comments to the Author” section, enter your conflict of interest statement in the “Confidential to Editor” section, and submit your "Accept" recommendation.

Reviewer #2: All comments have been addressed

Reviewer #3: All comments have been addressed

Reviewer #4: All comments have been addressed

2. Is the manuscript technically sound, and do the data support the conclusions?

Reviewer #2: Yes

Reviewer #3: Yes

Reviewer #4: Yes

3. Has the statistical analysis been performed appropriately and rigorously? 

Reviewer #2: Yes

Reviewer #3: Yes

Reviewer #4: Yes

4. Have the authors made all data underlying the findings in their manuscript fully available?

Reviewer #2: Yes

Reviewer #3: Yes

Reviewer #4: Yes

5. Is the manuscript presented in an intelligible fashion and written in standard English?

Reviewer #2: Yes

Reviewer #3: Yes

Reviewer #4: Yes

6. Review Comments to the Author

Reviewer #2: Thank you for revising the manuscript according to my comments. All concerns have been addressed well.

Reviewer #3: (No Response)

Reviewer #4: My comments were properly addressed in the previous submission. I have no addtional comments or concerns.

7. PLOS authors have the option to publish the peer review history of their article (what does this mean?). If published, this will include your full peer review and any attached files.

Reviewer #2: **Yes: **Satoshi Watanabe

Reviewer #3: **Yes: **Jiang Zhu

Reviewer #4: No

---

## [Editor Report · Acceptance letter]

9 Sep 2024

PONE-D-24-01682R2 

PLOS ONE

Dear Dr. Wislez, 

I'm pleased to inform you that your manuscript has been deemed suitable for publication in PLOS ONE. Congratulations! Your manuscript is now being handed over to our production team.

Kind regards, 

on behalf of

Dr. Fumihiro Yamaguchi 

Academic Editor

PLOS ONE